# Effect of Temperature on the Charge Transport Behavior of Epoxy/Nano−SiO_2_/Micro−BN Composite

**DOI:** 10.3390/nano12101617

**Published:** 2022-05-10

**Authors:** Fuqiang Tian, Jinmei Cao, Shuting Zhang

**Affiliations:** School of Electrical Engineering, Beijing Jiaotong University, Beijing 100044, China; 21117019@bjtu.edu.cn (J.C.); 19117036@bjtu.edu.cn (S.Z.)

**Keywords:** epoxy resin, nano–micro fillers, space charge, conduction current, high temperature

## Abstract

Thermally conductive epoxy resin composites are widely used as electrical equipment insulation and package materials to enhance heat dissipation. It is important to explore the dielectric properties of the composites at high temperatures for the safe operation of the equipment. This paper investigated the charge transport behavior of an epoxy/nano−SiO_2_/micro−BN composite at varied temperatures by combined analysis of the TSDC (thermally stimulated current), conduction current, complex permittivity and space charge distribution between 40 and 200 °C. The results show that ionic space charge accumulation was significantly suppressed in the composite at high temperatures. The conduction current increased gradually with temperature and manifested a remarkable shift from electron charge transport to ion charge transport near the glass transition temperature (*T*_g_). The real and imaginary permittivity showed an enormous increase above *T*_g_ for both the epoxy resin and the composite. The conduction current and permittivity of the composite were remarkably reduced in comparison to the epoxy resin. Therefore, the ionic process dominated the high temperature dielectric properties of the epoxy resin and the composite. The nano–micro fillers in the composite can significantly inhibit ion transport and accumulation, which can significantly enhance the dielectric properties of epoxy resin. Thus, the nano–micro composite has a strong potential application as a package material and insulation material for electronic devices and electrical equipment operated at high temperatures.

## 1. Introduction

With improvements in the integration degree and power density of electrical and electronic equipment, the heat generated per unit volume of equipment increases significantly. The continuous heat accumulation and the resulting temperature rise will accelerate the aging of the insulating dielectric and greatly reduce the reliability and life of the electrical and electronic equipment [1,2]. Therefore, the heat dissipation and high temperature dielectric properties of insulating materials are bottlenecks restricting the development of electrical equipment and electronic devices towards high power density and high integration [3].

Epoxy resin is widely used in electrical equipment insulation systems and electronic device packaging due to its excellent adhesion, corrosion resistance and dielectric properties [4,5,6,7,8]. However, during the operation of a device, the effect of high temperature and high electric field causes electrode electrons to be injected into the epoxy and become space charge. The space charge that accumulates in the medium will distort the local electric field and affect the breakdown characteristics of the materials, which leads to insulation deterioration and failure [9,10,11].

Additionally, the thermal conductivity of epoxy resin is only about 0.2 W/(m·K), which seriously affects the heat dissipation performance of the device. Some studies have shown that doping a certain amount of thermal conductive fillers in polymers can improve the thermal properties but adding too much filler will cause poor dielectric properties of the materials [12,13]. Therefore, developing new composite insulation materials which offer excellent thermal conductivity but low electrical conductivity has attracted significant interest worldwide [14,15]. In recent years, researchers have undertaken much research on the space charge and dielectric behavior of epoxy composites [16,17,18,19,20,21], but the charge transport and space charge accumulation behavior under high temperature and high electric field still need to be further explored.

In this paper, the charge transport behavior of an epoxy/nano−SiO_2_/micro−BN composite at varied temperatures is investigated by combined analysis of the TSDC (thermally stimulated current), conduction current, complex permittivity and space charge distribution between 40 and 200 °C.

## 2. Materials and Methods

### 2.1. Materials

Epoxy resin (diglycidyl ether of bisphenol−816B) was obtained from Mitsubishi Chemical (Tokyo, Japan). The weight ratio of curing agent (Hardener 113, Mitsubishi Chemical of Japan) to epoxy resin was 30 to 100. The fillers included SiO_2_ and BN particles. SiO_2_ was provided by Denka, Japan, which had an average particle size of 15 nm added to a ratio of 4 wt%. BN was purchased from Aerosil, Japan, in the shape of a flake with a diameter of 8 μm (D50), and was added to a volume ratio of 10 vol%. Volume ratio has a significant effect on improving thermal conductivity, so we used vol% for the ratio of the micro filler. 

### 2.2. Preparation of Epoxy/Nano−SiO_2_/Micro−BN Composite

The composite was prepared with the following procedure. First, mix SiO_2_ and BN fillers in a planetary centrifugal mixer (ARE−300, Thinky, Laguna Hills, CA, USA) for 3 min. Second, add the filler mixture to the epoxy resin and mix in the same mixer for 10 min. Third, evacuate the mixture in a vacuum oven at room temperature for 30 min in order to degas sufficiently. Fourth, mix the sufficiently degassed mixture with the hardener added into it for five minutes. Fifth, pour the previous mixture into a stainless steel mold and press tightly. Finally, precure it at 70 °C for three hours and postcure it at 120 °C for another three hours. Then, the sample obtained is naturally cooled down to room temperature. Hereafter, the thickness of the sample is 150–300 μm. 

### 2.3. Characterization

Scanning electronic microscopy (SEM): The dispersion of micro−sized and nano−sized fillers in the epoxy composite was studied using a thermal field emission scanning electron microscope (JSM−7100F, JEOL, Tokyo, Japan) to observe the cross−section of the sample after extraction with liquid nitrogen. The test voltage was 10 kV.

Complex permittivity measurement: The dielectric properties were measured using an impedance analyzer (126096, Solartron, Bognor Regis, UK). Samples of 200 μm thickness were operated at an AC voltage of 3 Vrms, at temperatures from 20 to 200 °C and at a frequency range from 10^−2^ to 10^6^ Hz. 

Conduction current measurement: The conduction current was measured using a Digital electrometer (Advantest R8252). The samples were operated in an atmosphere of nitrogen, at temperatures from 40 to 200 °C; the test voltage was a DC electric field (30 kV/mm). 

Space charge distribution measurement: The conducted currents under short−circuit, positive and negative electric field conditions were measured using a pulsed electroacoustic (PEA) device (Peanuts, Five Lab, Saitama, Japan) to study the space charge distribution of samples. The electric field was 10 kV/mm. 

Thermally stimulated discharge current measurement (TSDC): The Digital electrometer (Advantest R8252) was used to measure the thermally stimulated discharge current from 40 to 200 °C with a heating rate of 2 °C/min. 

At least two different samples of both the epoxy resin and the epoxy/nano−SiO_2_/micro−BN composite were used in all the tests to ensure the repeatability of the experimental results.

## 3. Results and Discussion

### 3.1. Subsection Nanoparticle Dispersion Characterization

The dispersion of the micro BN particles and SiO_2_ nanoparticles in the epoxy/nano−SiO_2_/micro−BN composite is shown in the SEM image in Figure 1. We can see from the SEM image that the BN particles are evenly distributed in the epoxy resin matrix. Most BN particle sizes are around 10 μm, with overall sizes ranging from 5 to 20 μm.

### 3.2. Space Charge Distribution

The spatial charge distribution in the epoxy resin polarized at test temperatures of 40 °C, 60 °C, 80 °C and 100 °C is shown in Figure 2. At 40 °C, it can be obviously seen that under the condition of short−circuit and the same or opposite direction biased conditions to the polarization electric field, negative homocharges accumulate alongside the cathode. In Figure 2a, it can be seen that (i) it is uncertain whether there is space charge around the anode when measured only under the short−circuited condition; (ii) when applying an electric field of the opposite polarity (negative DC), positive homocharges can be clearly seen around the anode; and (iii) a small amount of negative space charges can be observed near the anode. Thus, both electrons and holes of electrodes emptied into EP at 40 °C. The homocharges increased around both the cathode and the anode with the higher temperatures.

The space charge distributions in EP polarized at temperatures of 120, 140, 160, 180 and 200 °C are shown in Figure 3. Except for hole and electron injection, there was heterogeneous aggregation around the cathode with temperature, which was due to the migration and accumulation of mobile ions produced by the ionization of impurities. At this temperature range, the homopolar charge injected by the electrode is gradually covered up, and the impurity charge plays a dominant in space charge behavior. The maximum heterocharge density was 150 C/m^3^. 

The space charge distributions in epoxy/nano−SiO_2_/micro−BN composites polarized at temperatures from 40 to 200 °C are shown in Figure 4 and Figure 5. The space charge distribution and its dependence on temperature are very complicated due to the coexistence of homopolar and heterogenous charges [22]. Compared with epoxy resin, the density of heterocharges in the epoxy composite was smaller above 60 °C. Furthermore, the glass transition temperature (*T*_g_) of epoxy resin and the composite is between 100 and 120 °C, and the increase in heterocharges becomes much less significant when the temperature is above 100 °C. The homocharges injected by the electrode are just hidden by heterocharges above 120 °C [23].

Figure 6 shows the ion density of epoxy resin and the epoxy/nano−SiO_2_/micro−BN composite as a function of temperature from 40 to 200 °C. Among them, the ion density of the composite remains below 10 C/m^3^ all the time. However, the accumulated ionic space charges in epoxy resin are greater than those in the epoxy composite above 100 °C. Therefore, SiO_2_–BN fillers can significantly inhibit ion transport and accumulation in the epoxy/nano−SiO_2_/micro−BN composite.

### 3.3. Conduction Current

The conduction current behavior of epoxy resin and the epoxy/nano−SiO_2_/micro−BN composite with time at different temperatures varies, as shown in Figure 7. It shows how the conduction current behavior of epoxy resin and the epoxy/nano−SiO_2_/micro−BN composite changes over time at varied temperatures between 20 and 200 °C. From Figure 7a,b, it can be seen that the conduction current in both epoxy resin and the composite increases significantly when the temperature is greater than or equal to 100 °C. In order to explore the dependence of dielectric properties on temperature, the conductivity of epoxy resin and the composite was calculated using the steady−state conduction current and then by fitting the conductivity and temperature by the Arrhenius equation, as shown in Figure 8. The slope of the fitting curve changes significantly at around 100 °C, indicating that the conduction current or conductivity was dominated by different charge transport. It can be deduced that the conduction current temperatures and conductivity are dominated by ions at high temperatures and by electronic charges at low temperatures in epoxy resin, which also confirms that space charge, dielectric and conduction current behaviors at high temperatures are all ascribed to ion transport. 

Ions are transported through thermally activated hopping with a mobility that obeys the Arrhenius equation ($\mu =\mu_{0}e^{-E_{A}/RT}$), which describes the relationship between reactive activation energy, temperature and reactive rate. The activation energy (*E*_A_) of both epoxy resin and the epoxy/nano−SiO_2_/micro−BN composite is higher at 120–200 °C than 40–100 °C. The activation energy of the epoxy composite (0.9 eV) is lower than that of pure epoxy (1.20 eV) at a high temperature. The mobility of polymer molecular chains at the interface area in the composite is inhibited due to the steric hindrance effect of nano− and micro fillers, which equally introduces deep trapping sites at the interface area; thus, ionic charge transport in the composite is suppressed compared to the pure polymer [24,25].

### 3.4. Dielectric Behavior

Figure 9 shows the complex permittivity spectra of epoxy resin and the composite at different temperatures. In particular, both the real part (ε_r_′) and the imaginary part (ε_r_″) of the complex permittivity of epoxy resin and the composite increase significantly with temperature. This shows that at a higher temperature and lower frequency, ions move to the electrode interface, and the polarization of the electrode induces a large amount of mirror charge, which causes a large number of ions to gather around the electrode; thus the ε_r_′ and ε_r_″ increase to high values. Furthermore, it has verified that the glass transition temperature is about 100 °C, and the increase in ε_r_′ is due to the accumulation of impurity ions near the electrode. 

When the test temperature of epoxy and the composite material is higher than the glass transition temperature, the molecular chain movement in the amorphous region of the material is more active, which significantly improves the charge transfer ability. The charge can easily transfer to the other side of the electrode, forming electrode polarization. However, due to the barrier between the electrode and dielectric interface, charge extraction is limited and heteropolar space charge is formed. The higher the temperature of heteropolar space charge is, the more ions are activated. Therefore, electrode polarization and ion charge transport at high temperatures can make the complex dielectric constant of the material larger. There are two main reasons for the generation of ions in epoxy resin: first, a small amount of sodium ions and chloride ions remain in the synthesis process of epoxy resin in an alkaline environment; second, polar impurities and unreacted parts of the curing agent are ionized at high temperatures. An obvious relaxation process occurs in the spectra when the ambient temperature of epoxy and the composite is lower than the glass transition temperature. It is mainly because the relaxation process is triggered by the orientation of molecular chain side chains and side groups, and the relaxation process moves to a higher frequency with temperature. 

The changes in the complex dielectric constants of epoxy resin and the composite at 0.01 and 0.1 Hz with temperature are shown in Figure 10. It can be seen that the ε_r_′ and ε_r_″ of epoxy and the composite increase significantly with the temperature above the glass transition temperature, and compared with the epoxy resin, the ε_r_′ and ε_r_″ of the composite are lower. This is because the nano and micro fillers in the composite material fill the gap between the molecular chains and hinder the movement of the molecular chains, resulting in ion transport becoming difficult when the test temperature is above *T*_g_. This conclusion is further confirmed by the fact that the SiO_2_–BN fillers can significantly inhibit ion transport and accumulation in the epoxy composite at high temperatures.

### 3.5. TSDC (Thermally Stimulated Discharge Current)

Figure 11 shows the TSDC of epoxy resin and the epoxy/nano−SiO_2_/micro−BN composite as a function of temperature from 20 to 200 °C. By comparing the TSDC spectra of epoxy resin and the composite, it can be known that the current peak in both materials appears at 80–110 °C. Furthermore, when the temperature is above 140 °C, the current peak is ascribed to the ion transport.

Figure 12 shows the α relaxation peak of the TSDC spectra of epoxy resin and the epoxy/nano−SiO_2_/micro−BN composite. For epoxy resin, the peak current increases from 80 to 120 °C, reaches the maximum at 120 and 140 °C and then decreases from 140 to 200 °C. For the epoxy composite, the peak current values are nearly the same for all polarization temperatures, and the current peak moves to a high temperature with the increase in *T*_p_, indicating a dispersion of the α relaxation process. In addition, both the peak current and peak temperature of the epoxy composite are higher than those of epoxy resin, indicating that it is more difficult for ions to transport in the composite, which is also confirmed by the conductivity.

## 4. Conclusions

The space charge distribution, conduction current, complex permittivity and TSDC of epoxy resin and an epoxy/nano−SiO_2_/micro−BN composite under varied temperatures have been investigated. The results suggest that the space charge behavior in epoxy resin and the composite is dominated by electronic charge accumulation below the glass transition temperature (*T*_g_) and by ion accumulation at temperatures above the *T*_g_. The enormous increase in the complex dielectric permittivity of both epoxy resin and the composite is due to electrode polarization caused by ion charge transport above the *T*_g_. However, SiO_2_–BN nano–micro fillers can significantly inhibit ion transport and accumulation by hindering the movement of the molecular chains, thus leading to a lower conduction current and higher ρ peak temperature in TSDC spectra.

In summary, the nano–micro fillers have a steric hindrance effect on epoxy chain segment movement and lead to limited ion transport. The epoxy/nano−SiO_2_/micro−BN composite exhibits remarkably improved dielectric properties at high temperatures compared to epoxy resin and is more suitable for the insulation materials in equipment and devices operating at high temperatures.

## Figures and Tables

**Figure 1 nanomaterials-12-01617-f001:**
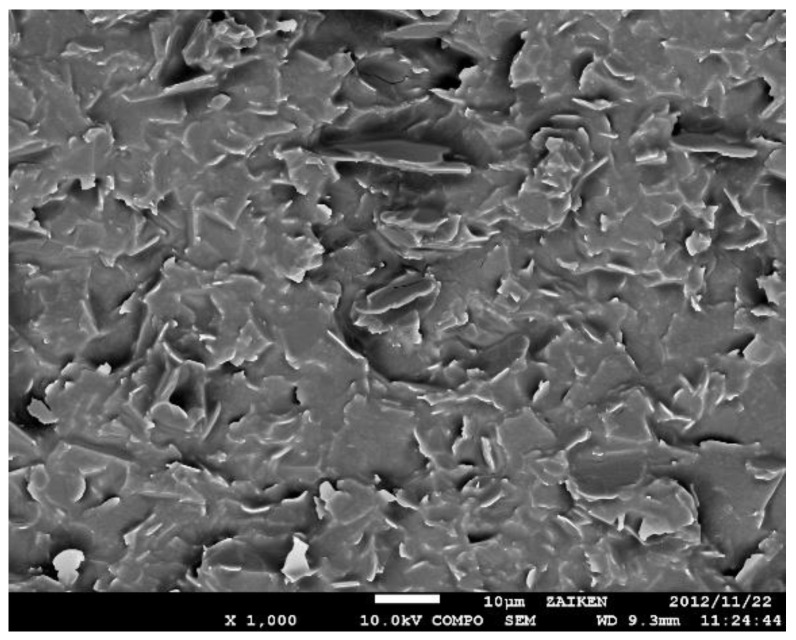
SEM section morphology graph of epoxy/nano−SiO_2_/micro−BN composite.

**Figure 2 nanomaterials-12-01617-f002:**
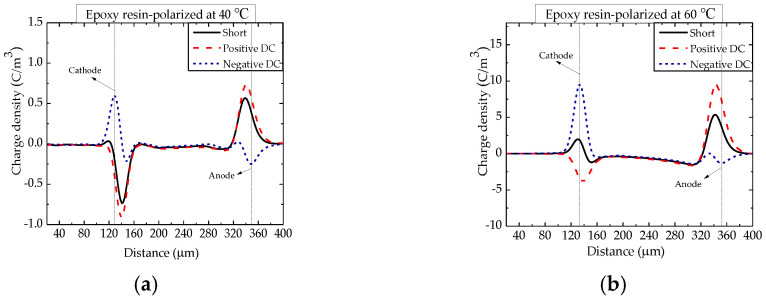
Space charge distribution in epoxy resin polarized at temperatures between 40 and 100 °C. (**a**) 40 °C; (**b**) 60 °C; (**c**) 80 °C and (**d**) 100 °C.

**Figure 3 nanomaterials-12-01617-f003:**
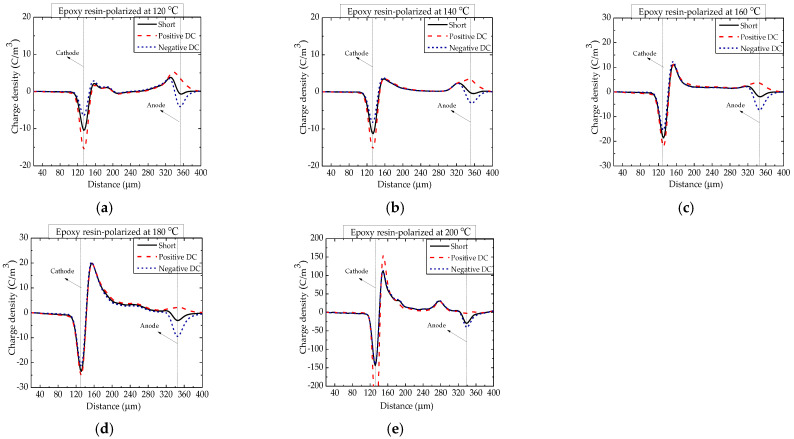
Space charge distribution in epoxy resin polarized at temperatures between 120 and 200 °C. (**a**) 120 °C; (**b**) 140 °C; (**c**) 160 °C; (**d**) 180 °C and (**e**) 200 °C.

**Figure 4 nanomaterials-12-01617-f004:**
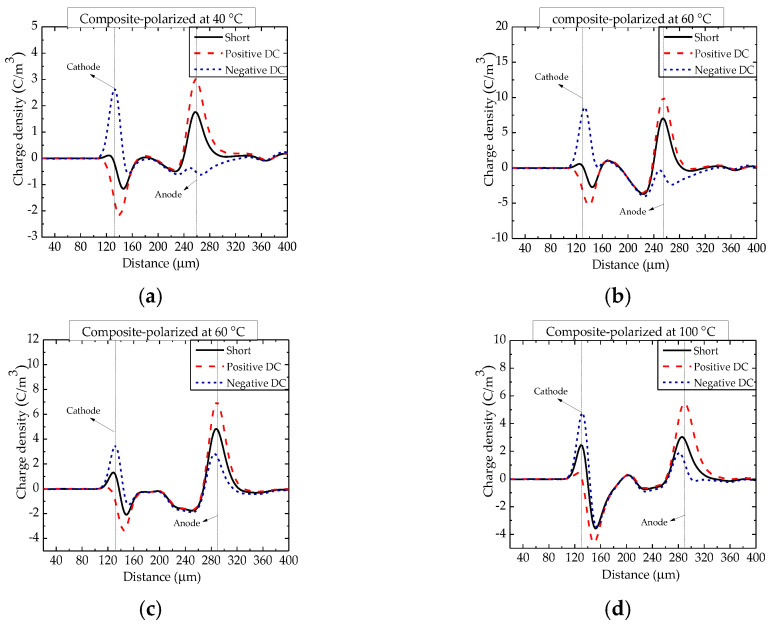
Space charge distribution in epoxy/nano−SiO_2_/micro−BN composite polarized at temperatures between 40 and 100 °C. (**a**) 40 °C; (**b**) 60 °C; (**c**) 80 °C and (**d**) 100 °C.

**Figure 5 nanomaterials-12-01617-f005:**
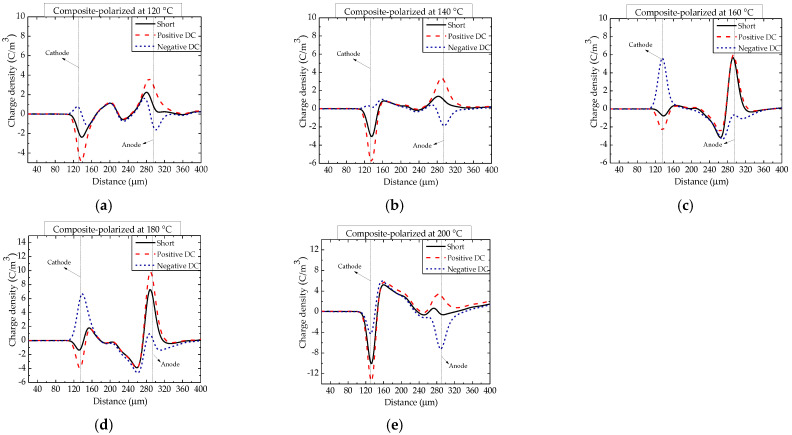
Space charge distribution in epoxy/nano−SiO_2_/micro−BN composite polarized at temperatures between 120 and 200 °C. (**a**) 120 °C; (**b**) 140 °C; (**c**) 160 °C; (**d**) 180 °C and (**e**) 200 °C.

**Figure 6 nanomaterials-12-01617-f006:**
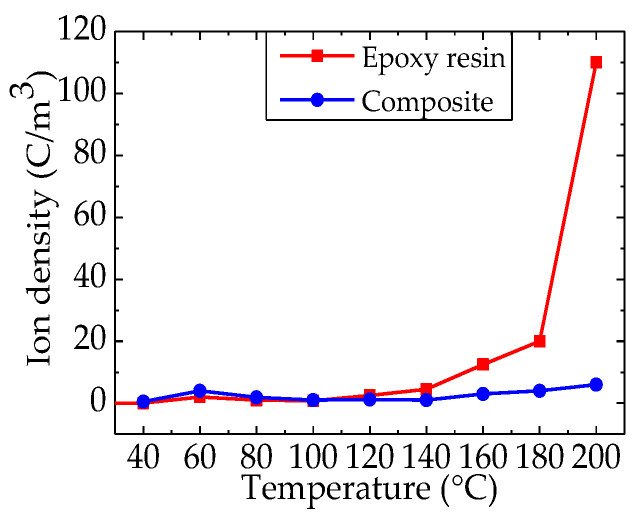
Ion density of epoxy resin and epoxy/nano−SiO_2_/micro−BN composite at varied temperatures.

**Figure 7 nanomaterials-12-01617-f007:**
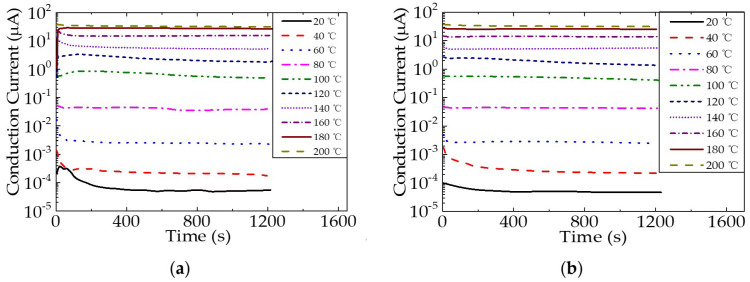
Conduction current of epoxy resin (**a**) and epoxy/nano−SiO_2_/micro−BN composite (**b**) as a function of time.

**Figure 8 nanomaterials-12-01617-f008:**
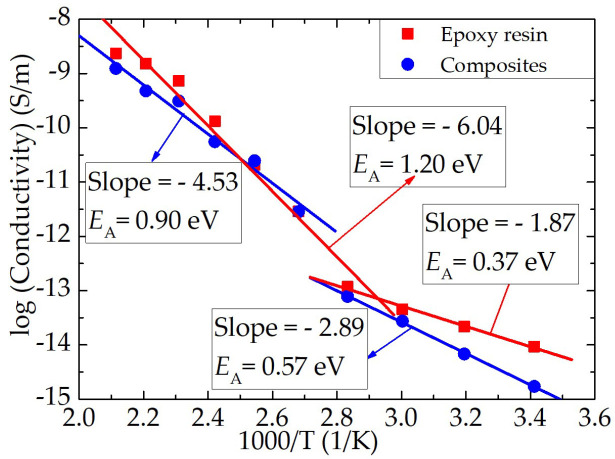
Arrhenius fitting of the conductivity calculated from the conduction current. *E*_A_: activation energy. Solid lines are the fitted curves.

**Figure 9 nanomaterials-12-01617-f009:**
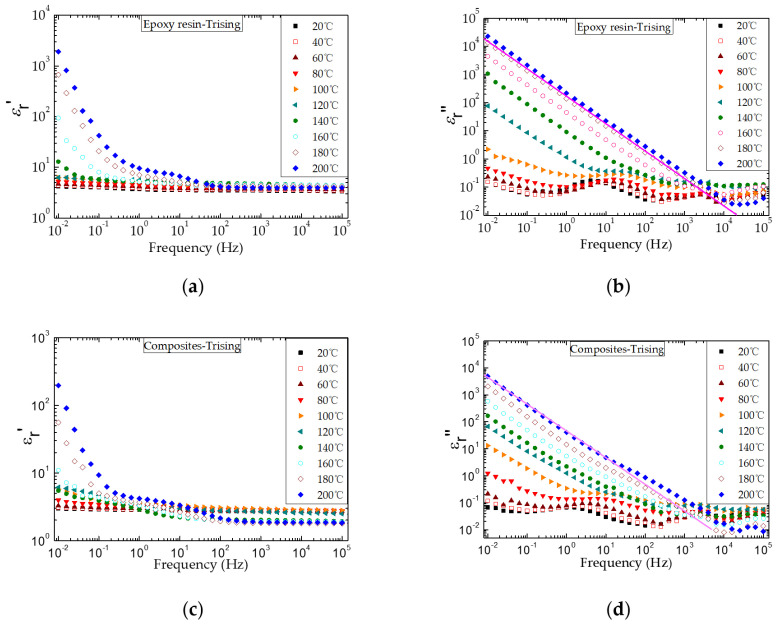
Real part (ε_r_′) and imaginary part (ε_r_″) of the complex permittivity of epoxy resin (**a**,**b**) and the composite (**c**,**d**) at different temperatures.

**Figure 10 nanomaterials-12-01617-f010:**
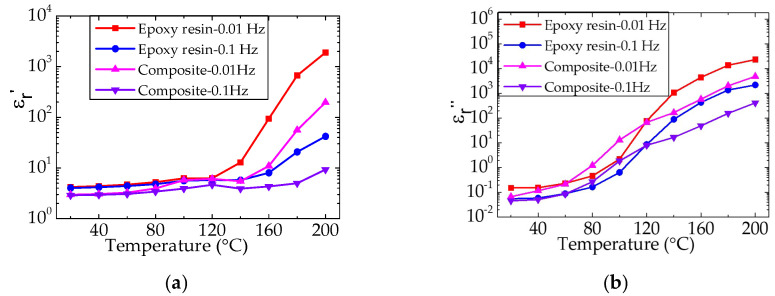
Real part (ε_r_′) and imaginary part (ε_r_″) of the complex permittivity of epoxy resin (**a**) and its composite (**b**) at different temperatures.

**Figure 11 nanomaterials-12-01617-f011:**
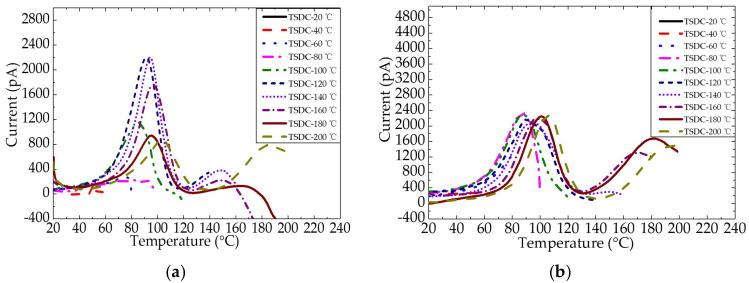
TSDC of epoxy resin (**a**) and epoxy/nano−SiO_2_/micro−BN composite (**b**).

**Figure 12 nanomaterials-12-01617-f012:**
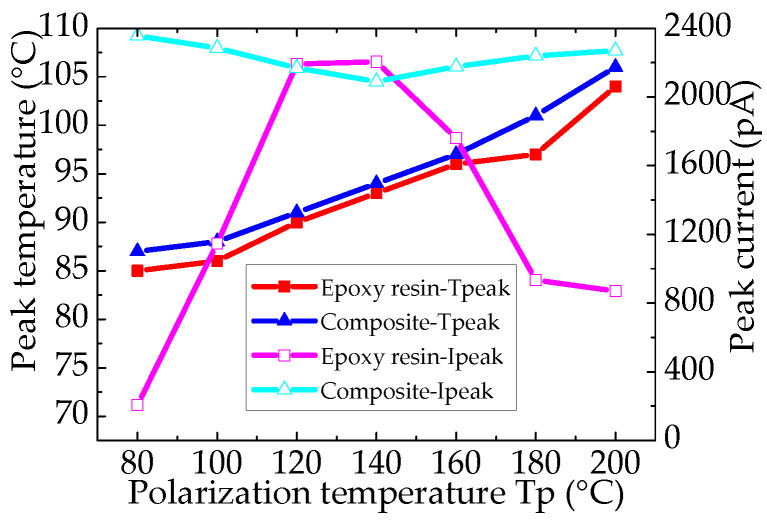
Comparison of α relaxation peak of TSDC spectra.

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
