# Peer review of "Effect of Temperature on the Charge Transport Behavior of Epoxy/Nano−SiO2/Micro−BN Composite"

_nanomaterials, 2022, doi:10.3390/nano12101617_

Round 1

Reviewer 1 Report

I would like the authors to put more effort and fix the below or provide more information about their research. I recommend publishing the manuscript only after revision and their responses to the last two questions below:

  • The expression: Scanning Electironic Microscope (ESM): in page 2 line 74 needs to be modified.
  • epsilon_r' and  epsilon_r" have not been typed correctly in lines 194 to 196.
  • What is the Arrhenius? Is that the formula for the temperature dependence of reaction rates? The authors could introduce a reference about it for a beginner.
  • Accurate acquiring of parameters of charge transport processes is difficult. In most cases, they are only applicable to the specified experimental condition, even for the same material, the parameters may alter owing to the difference of experimental condition. Therefore, a question raises. How many samples did you prepare for measurements? Are the reported values average values?
  • the ionization process of impurity particles in the material is important for the charge transport properties of materials under strong electric field and high temperature. Could the authors explain how they did consider this effect?

Author Response

Dear Editor,

We would like to thank you for your kind letter and all the reviewers’ constructive comments. These comments are all valuable and helpful for improving this manuscript entitled “Effect of Temperature on the Charge Transport Behavior of Epoxy/nano-SiO2/micro-BN Composite (ID: nanomaterials-1700154)”. According to the reviewers’ comments, we have modified the manuscript to meet the requirements. Modifications are marked with red color in the manuscript. Point-by-point responses to the reviewers are listed below in this letter. We would like to express our sincere thanks to the reviewers and you for the constructive and positive comments.

Point 1: The expression: Scanning Electironic Microscope (ESM): in page 2 line 74 needs to be modified.

epsilon_r' and epsilon_r" have not been typed correctly in lines 194 to 196.

Response 1: Scanning Electironic Microscope (ESM) have be modified as Scanning Electronic Microscope (SEM).

epsilon_r' and epsilon_r" have be modified as εr' and εr''.

Point 2: What is the Arrhenius? Is that the formula for the temperature dependence of reaction rates? The authors could introduce a reference about it for a beginner.

Response 2: We made several amendments in page 6 lines 171 to 172, shown below.

“Ions are transported through thermally activated hopping with a mobility that obeys the Arrhenius equation (), it describes the relationship between reactive activation energy, temperature and reactive rate.”

Point 3: Accurate acquiring of parameters of charge transport processes is difficult. In most cases, they are only applicable to the specified experimental condition, even for the same material, the parameters may alter owing to the difference of experimental condition. Therefore, a question raises. How many samples did you prepare for measurements? Are the reported values average values?

Response 3: In order to confirm the repeatability of the experimental results, at least two different samples are used for all the tests for each kind of the material. In addition, a great number of groups of the TSDC, conduction current, complex permittivity and space charge distribution data were measured from 40 to 200 °C, we mainly intend to compare and analyse the evolution rules and laws of the charge transport behavior of epoxy resin and its composite at different temperature, so the deviation of the measured result won’t change the main conclusion.

Point 4: the ionization process of impurity particles in the material is important for the charge transport properties of materials under strong electric field and high temperature. Could the authors explain how they did consider this effect?.

Response 4: There are two main reasons for the generation of ions in epoxy resin: a small amount of sodium ions and chloride ions remain in the synthesis process of epoxy resin in alkaline environment; the unreacted hardener and nonreactive polar impurities in the hardener would be ionized at elevated temperatures. The molecular chain movement in the amorphous region of the material is more active at high temperature and lower frequency, which significantly improves the charge transfer ability. The charge can easily transfer to the other side of the electrode, forming the electrode polarization. ions move to the electrode interface, and the polarization of the electrode induces a large amount of mirror charge, which causes a large number of ions to gather around the electrode.

If you have any queries, please don’t hesitate to contact us.

Thank you and best regards.

Yours sincerely,

Jinmei Cao

Reviewer 2 Report

This manuscript presents thermally conductive epoxy resin composites filled with BN and silica nanoparticles. I have some comments that should be addressed:

  1. In introduction, thermal conductivity was discussed, and the authors should motivate this is important feature even for HVDC insulation to have high thermal conductivity while lower electrical conductivity is required. These articles are beneficial to use:

High Voltage 2022, 7(2):251-259

High Voltage 2020,5(3),280-286

  1. The authors should highlight as outlook, will focus on thermal conductivities.
  2. The authors missed the discussion of interface in polymer nanocomposites. The trapping of charges occurred at the interface of nanoparticles and additives. Please clarify this concept and mechanisms. These papers are important:

Advanced Materials 2021, 33 (27), 2100714

Advanced Materials 2018, 30 (4), 1703624

  1. There is a very similar investigation on this topic (Energies 2021, 14, 4645.), What is the reflection of authors? What are the novelties in this work?
  2. I suggest rewriting the abstract and conclusion, and instead of raw results the authors should discuss what is the added value of this research.

Author Response

Response to Reviewer 2 Comments

Dear Editor,

We would like to thank you for your kind letter and all the reviewers’ constructive comments. These comments are all valuable and helpful for improving this manuscript entitled “Effect of Temperature on the Charge Transport Behavior of Epoxy/nano-SiO2/micro-BN Composite (ID: nanomaterials-1700154)”. According to the reviewers’ comments, we have modified the manuscript to meet the requirements. Modifications are marked with red color in the manuscript. Point-by-point responses to the reviewers are listed below in this letter. We would like to express our sincere thanks to the reviewers and you for the constructive and positive comments.

Point 1: In introduction, thermal conductivity was discussed, and the authors should motivate this is important feature even for HVDC insulation to have high thermal conductivity while lower electrical conductivity is required. These articles are beneficial to use:

High Voltage 2022, 7(2):251-259

High Voltage 2020,5(3),280-286

Response 1: We made several amendments to the introduction in page 2 lines 53 to 57, shown below.

“Some studies have shown that doping a certain amount of thermal conductive fillers in polymers can improve the thermal properties, but adding too much fillers will cause poor dielectric properties of materials [12,13]. Therefore, developing new composite insulation materials which offer excellent thermal conductivity while lower electrical conductivity has attracted significant interest worldwide[15,16].”

The above two articles have be cited.

Point 2: The authors should highlight as outlook, will focus on thermal conductivities.

The authors missed the discussion of interface in polymer nanocomposites. The trapping of charges occurred at the interface of nanoparticles and additives. Please clarify this concept and mechanisms. These papers are important:

Advanced Materials 2021, 33 (27), 2100714

Advanced Materials 2018, 30 (4), 1703624

Response 2: We made several amendments in page 6 lines 176 to 180, shown below.

“The mobility of polymer molecular chains at the interface area in the composites are inhibited due to the steric-hindrance effect of nano and micro fillers, which equivalently introduces deep trapping sites at the interface area and thus ionic charge transport in the composite are suppressed compared to the pure polymer[24,25].”

The above two articles have be cited.

Point 3: There is a very similar investigation on this topic (Energies 2021, 14, 4645.), What is the reflection of authors? What are the novelties in this work?

Response 3: Despite the fillers of two articles are both BN and SiO2, the polymer of article entitled “Numerical Simulation on Charge Transport and DC Breakdown in Polyethylene-Based Micro-h-BN/Nano-SiO2 with Filler Orientation Dependent Trap Energy” is polyethylene, and it mainly investigates the effect of orientation of micro-h-BN on charge transport and DC breakdown of PE-based micro/nano-composites. The polymer used in our manuscript is epoxy resin, long-term operation of electronic devices at high temperature and high electric field will cause aging of insulation, which will seriously affect the reliability and service life of the components. Therefore, the charge transport and dielectric behavior of epoxy resin and its composite under varied temperatures are mainly studied in this paper.

Point 4: I suggest rewriting the abstract and conclusion, and instead of raw results the authors should discuss what is the added value of this research.

Response 4: We made several amendments to the abstract, shown below.

“Thermally conductive epoxy resin composites are widely used as electrical equipment insulation and package materials to enhance the heat dissipation. It is important to explore the dielectric properties of the composite at high temperatures for the safe operation of the equipment. This paper investigated the charge transport behavior of epoxy/ nano-SiO2/micro-BN composite at varied temperatures by combined analysis of the TSDC (thermally stimulated current), conduction current, complex permittivity and space charge distribution between 40 and 200 °C. The results show that ionic space charge accumulation is significantly suppressed in the composite at high temperatures. The conduction current increases gradually with temperature and manifests a re-markable shift from electron charge transport to ion charge transport near the glass transition temperature (Tg). The real and imaginary permittivity shows an enormous increase above Tg for both the epoxy resin and the composite. The conduction current and permittivity of the composite is remarkably reduced in comparison to the epoxy resin. Therefore, ionic process dominates the high temperature dielectric properties of the epoxy resin and the composite. The nano - micro fillers in the composite can significantly inhibit ion transport and accumulation, which can significantly enhance the dielectric properties of epoxy resin. Thus, the nano-micro composite has a strong potential application as package materials and insulation materials for electronic devices and electrical equipment operated at high temperatures.”

We made several amendments to the conclusion, shown below.

“Space charge distribution, conduction current, complex permittivity and TSDC of epoxy resin and epoxy/nano-SiO2/micro-BN composite under varied temperatures have been investigated. The results suggest that the space charge behavior in epoxy resin and the composite is dominated by electronic charge accumulation below glass transition temperature (Tg), and by ion accumulation at temperatures above Tg. The enormous increase of complex dielectric permittivity of both epoxy resin and the composite are due to electrode polarization caused by ion charge transport above Tg. Whereas SiO2-BN nano-micro fillers can significantly inhibit ion transport and accu-mulation by hinder the movement of the molecular chains, which leading to lower conduction current, higher ρ peak temperature in TSDC spectra.

In summary, the nano-micro fillers have a steric hindrance effect on epoxy chain segment movement and lead to limited ion transport, the epoxy/nano-SiO2/micro-BN composite exhibits remarkably improved dielectric properties at high temperatures compared to epoxy resin and is more suitable for the insulation materials in equipment and devices operating at high temperatures.”

If you have any queries, please don’t hesitate to contact us.

Thank you and best regards.

Yours sincerely,

Jinmei Cao

Reviewer 3 Report

May the authors comment the morphology of the materials? Is this important for the applicability of the materials under study? Do the materials undergoo any morphology changes at the application conditions?

Author Response

Response to Reviewer 3 Comments

Dear Editor,

We would like to thank you for your kind letter and all the reviewers’ constructive comments. These comments are all valuable and helpful for improving this manuscript entitled “Effect of Temperature on the Charge Transport Behavior of Epoxy/nano-SiO2/micro-BN Composite (ID: nanomaterials-1700154)”. According to the reviewers’ comments, we have modified the manuscript to meet the requirements. Modifications are marked with red color in the manuscript. Point-by-point responses to the reviewers are listed below in this letter. We would like to express our sincere thanks to the reviewers and you for the constructive and positive comments.

Point 1: May the authors comment the morphology of the materials? Is this important for the applicability of the materials under study? Do the materials undergo any morphology changes at the application conditions?

Response 1: The epoxy composite prepared in this paper is a thermoset resin and thus its morphology does not change obviously after heating.

If you have any queries, please don’t hesitate to contact us.

Thank you and best regards.

Yours sincerely,

Jinmei Cao

Round 2

Reviewer 2 Report

The authors addressed the given comments.